# DynamicBench: Evaluating Real-Time Report Generation in Large Language Models

## Abstract

Traditional benchmarks for large language models (LLMs) typically rely on static evaluations through storytelling or opinion expression, which fail to capture the dynamic requirements of real-time information processing in contemporary applications. To address this limitation, we present DynamicBench, a benchmark designed to evaluate the proficiency of LLMs in storing and processing up-to-the-minute data. DynamicBench utilizes a dual-path retrieval pipeline, integrating web searches with local report databases. It necessitates domain-specific knowledge, ensuring accurate responses report generation within specialized fields. By evaluating models in scenarios that either provide or withhold external documents, DynamicBench effectively measures their capability to independently process recent information or leverage contextual enhancements. Additionally, we introduce an advanced report generation system adept at managing dynamic information synthesis. Our experimental results confirm the efficacy of our approach, with our method achieving state-of-the-art performance, surpassing GPT4o in document-free and document-assisted scenarios by 7.0% and 5.8%, respectively. The code and data will be made publicly available.

## 1 Introduction

In recent years, Large Language Models (LLMs) have revolutionized natural language processing, displaying exceptional proficiency in tasks ranging from language generation to contextual comprehension across various domains. However, traditional benchmarks remain confined to static evaluations, often relying on storytelling or expression of opinion. Such static, subjective assessment criteria fail to capture the dynamic nature of real-time information processing, which is crucial for understanding the true capabilities of LLMs (Wu et al., 2025; Que et al., 2024).

Addressing these limitations, we introduce DynamicBench, a benchmark designed to evaluate LLMs' proficiency in acquiring and processing real-time data. Distinguished by its demand for contemporary information retrieved through web searches and database queries, DynamicBench necessitates that models possess the most up-to-date knowledge for accurate responses. Utilizing a dual-path retrieval pipeline, DynamicBench combines local report databases with web searches, ensuring access to comprehensive data for thorough report evaluation. DynamicBench assesses a wide array of domains, capturing the latest dynamics across critical categories such as *Tech & Science*, *Economy & Environment*, *Culture & Health*, and *International & Politics*. Through both scenarios, providing or withholding external documents, DynamicBench evaluates a model's capability to store knowledge or process recent external information effectively. This requirement for precise data collection within specialized fields guarantees the accuracy and objectivity of the evaluation process, bridging the gap in current methodologies regarding objective and real-time assessments.

Beyond the benchmark itself, our contribution includes a robust solution for report generation, adept at tackling the complex challenges posed by dynamic information generation. Our system begins with report planning based on the query followed by query generation and resource aggregation using a dual-path retrieval pipeline from both local and online data. The system self-assesses whether further information gathering is necessary and ensures adequate information collection, informing detailed report writing that integrates tables and charts for enhanced clarity. Ultimately, it outputs a comprehensive, coherent report that reflects the latest data. Experimental results demonstrate the efficacy of our methods. We evaluate LLMs under two conditions: without and with docu-

Figure 1: Query examples across four major categories: *Tech & Science*, *Economy & Environment*, *Culture & Health*, and *International & Politics*, each with multiple subcategories.

ment assistance, and analyze their performance across different domains in both scenarios. Our approach showcases state-of-the-art performance across several metrics, surpassing GPT4o by 7.0% and 5.8%, respectively.

In summary, our contributions are as follows:

1. We introduce DynamicBench, a novel benchmark that evaluates LLMs based on real-time information acquisition and processing capabilities, utilizing a dual-path retrieval system that combines local and online data sources.

2. We develop a comprehensive report generation system that plans, searches, and writes detailed reports, ensuring the integration of up-to-date information for accurate and coherent documentation.

3. We demonstrate through experimental results the advanced capabilities of our approach, which achieves state-of-the-art performance compared to leading LLMs, highlighting significant improvements across multiple metrics.

## 2 RELATED WORKS

**Writing Benchmarks.** Recent advancements in evaluating Large Language Models (LLMs) have led to the creation of several benchmarks aimed at assessing different aspects of language generation and comprehension. LongBench-Write (Bai et al., 2024) focuses on understanding model capabilities in adhering to complex writing tasks within LLMs. HelloBench (Que et al., 2024) expands evaluation efforts by categorizing long text generation into distinct tasks such as open-ended QA and heuristic text generation. EQ-Bench (Paech, 2024) introduces an evaluation of emotional intelligence by assessing LLMs' abilities to comprehend and predict emotional intensities in dialogues. WritingBench (Wu et al., 2025) offers a comprehensive evaluation across domains and subdomains, including creative and technical writing. These traditional methods which predominantly focused on storytelling or opinion expression, adopting static and subjective evaluation criteria. In comparison, our system not only offers a holistic framework that covers a wide range of topics and evaluates various aspects of writing, but also utilizes real-time web searches and database queries to access the latest information. Thus, our system evaluates models' ability to process and utilize real-time information effectively. Moreover, our benchmark necessitates constructing precise reports within specialized fields, thus ensuring the accuracy and objectivity of the information utilized. These attributes enable our benchmark to bridge the gap in the current benchmarks concerning objective and real-time assessments.

**Long-Context Capabilities of LLMs.** Large Language Models (LLMs) such as Claude-3 (Anthropic, 2023), DeepSeek-R1 (DeepSeek-AI, 2025), DeepSeek-v3 (DeepSeek-AI et al., 2025), GPT-4o (OpenAI et al., 2024), and Qwen-2.5 (Qwen et al., 2025) have demonstrated remarkable capabilities in various domains, including understanding and generating complex language tasks. These

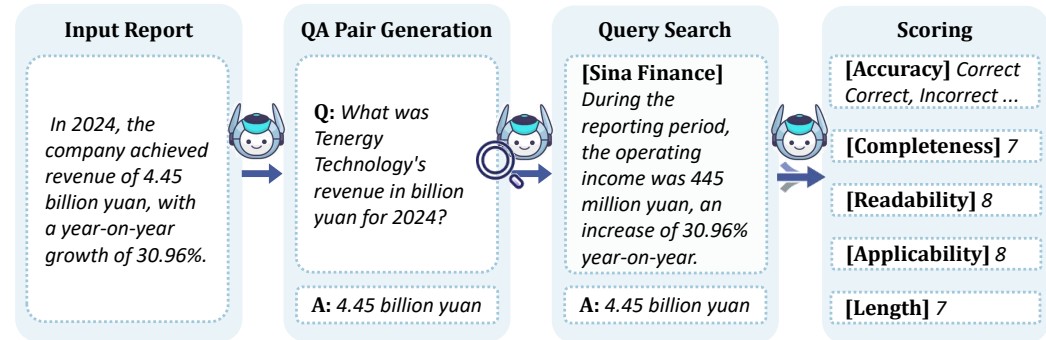

Figure 2: The evaluation system process begins with the generation of question and answer (Q&A) pairs from key details extracted from the input report, which are used as queries in a dual-path retrieval strategy. This strategy involves searching both online and within a local financial report database to gather relevant information. The system assesses the accuracy of each Q&A pair by aligning reported data with retrieved information and calculates the *accuracy*. The *completeness*, *readability*, *applicability*, and *length* of the report are also evaluated based on the retrieved information.

models serve as foundational tools for numerous applications, yet often face limitations in generating extended outputs or adhering to intricate task constraints. LongWriter (Bai et al., 2024) addresses the output length limitation in current LLMs by proposing AgentWrite, an agent-based pipeline that enables models to generate coherent outputs exceeding 20,000 words. Suri (Pham et al., 2024) introduces a multi-constraint instruction-following approach for generating long-form texts. It can generate significantly longer texts with sustained quality and compliance to constraints. In contrast, our work surpasses previous efforts by effectively generating extended content with enhanced coherence and quality.

## 3 METHODOLOGY

In order to address the challenges posed by the dynamic information generation and the need for accurate report construction, our methodology centers around the development of a benchmark and a robust system solution. In Sec. 3.1, we introduce a benchmark is designed to assess the ability of LLMs in acquiring and processing real-time data. In Sec. 3.2, we propose our report generation system solution.

### 3.1 DYNAMICBENCH

Traditionally, benchmarks (Wu et al., 2025; Que et al., 2024) have relied on storytelling or opinion expression, which are non-time-sensitive due to their static nature. In contrast, our benchmark, as exemplified in Fig. 1, requires contemporary, time-sensitive information retrieved via web search and database queries. This approach necessitates the possession of the most up-to-date domain-specific knowledge for accurate responses, thus assessing the capability of current models in acquiring and processing real-time information. Moreover, unlike traditional subjective evaluations, our benchmark demands the collection of data to construct reports within specialized fields, ensuring the accuracy and objectivity of the evaluation utilized. These attributes position our benchmark to narrow the gap in the current benchmarks regarding objective and real-time assessments. Our benchmark comprises the following categories:

1. **Tech & Science**: *technology* and *science*.

2. **Economy & Environment**: *economy* and *environment*.

3. **Culture & Health**: *society and culture*, *health*, and *sports*.

4. **International & Politics**: *international relations* and *law and politics*.

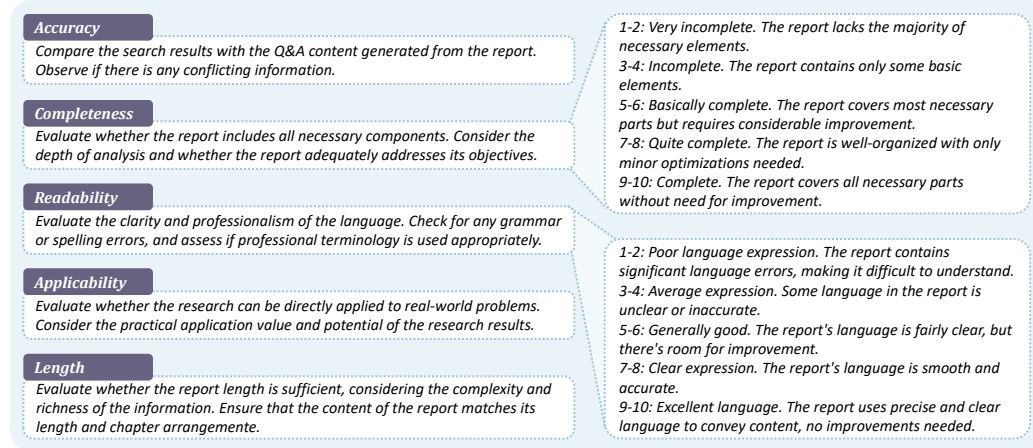

Figure 3: Evaluation criteria for report analysis, focusing on five distinct metrics: *accuracy*, *completeness*, *readability*, *applicability*, and *length*. Each criterion is supported by a detailed description to guide evaluators in comparing search results, examining the depth of analysis, assessing linguistic clarity and professionalism, evaluating real-world applicability, and verifying the adequacy of report length relative to content richness. The right panels examplify rating scale from 1 to 10 for *completeness* and *readability*, offering specific guidelines on how each score reflects the report's quality and coherence.

### 3.1.1 INFORMATION RETRIEVAL PROCESS

To construct our local report database, we sourced 148,589 annual reports from 10,338 global companies from AnnualReport[1]. These reports cover a diverse array of domains including economy, environment, technology, science, culture, health, laws, politics, etc. We leverage a retrieval-augmented generation (RAG) approach to perform information retrieval, as illustrated in Fig. 4.

The process involves using a context encoder to encode the local report database and a query encoder to encode incoming queries. Each report and query is transformed into embeddings, denoted as $\mathbf{E}_c$ for context and $\mathbf{E}_q$ for queries. The similarity between these embeddings is computed using cosine similarity, defined as:

$$\text{Similarity}(\mathbf{E}_c, \mathbf{E}_q) = \frac{\mathbf{E}_c \cdot \mathbf{E}_q}{\|\mathbf{E}_c\|\|\mathbf{E}_q\|} \tag{1}$$

The system effectively extracts the report block with the highest similarity score as the most relevant information. This process is mathematically represented as selecting the block $\mathbf{B}^*$ such that:

$$\mathbf{B}^* = \text{argmax}_i \ \text{Similarity}(\mathbf{E}_{c_i}, \mathbf{E}_q) \tag{2}$$

We utilize a dual-path retrieval pipeline, obtaining information through both the local report database and web searches. This comprehensive approach ensures that our system leverages the available data for robust and comprehensive report generation.

### 3.1.2 EVALUATION PROCESS

As illustrated in Fig. 2. The initial stage of our evaluation process involves extracting key information from the input report to generate question-and-answer (Q&A) pairs. These pairs form the basis for subsequent information retrieval, wherein queries derived from these pairs are employed within a dual-path retrieval strategy. This strategy utilizes both web searches and the local financial report database to gather comprehensive information. Once retrieved, our system evaluates several metrics, as depicted in Fig. 3. These metrics include:

---

[1]https://www.annualreports.com

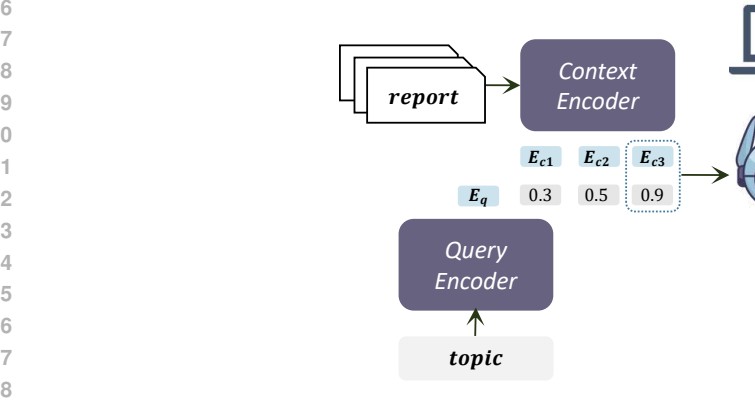

Figure 4: Dual-Path Retrieval Report Generation System that combines retrieval-augmented generation (RAG) from a local financial report database and Web Search to gather information. The related information are fed into a LLM for comprehensive report generation.

1. **Accuracy.** The accuracy of each Q&A pair is determined by assessing the alignment between reported data and the information retrieved; If the search data corroborates the Q&A or no discrepancies are found, the system labels it as *Correct*. Conversely, if relevant content is missing or conflicts are detected, it may be marked as *Cannot Determine* or *Incorrect*. The average accuracy across all queries is calculated to determine the final accuracy metric of the report.

2. **Completeness.** This metric assesses whether the report includes all necessary elements and adequately addresses its objectives. Evaluators use a rating scale to determine completeness, from very incomplete (1-2 points) to fully complete (9-10 points).

3. **Readability.** This criterion evaluates the clarity and professionalism of the report's language, checking for grammatical and spelling errors, as well as the appropriate use of professional terminology. Readability is rated from poor language expression (1-2 points) to excellent language use (9-10 points).

4. **Applicability.** This metric gauges the practical application value of the research findings, assessing whether the report can directly contribute to solving real-world problems. Applicability is ranked from poor applicability (1-2 points) to significant application value (9-10 points).

5. **Length.** This criterion evaluates if the report's length sufficiently covers the complexity and richness of the information presented. Length is rated from highly insufficient (1-2 points) to perfectly sufficient (9-10 points), considering the adequacy of each chapter's content.

## 3.2 REPORT GENERATION PROCESS

In addressing complex report generation challenges, our system provides a structured methodology for high-quality output, as depicted in Fig. 5.

1. **Section Planning.** This initial phase involves the establishment of major section titles based on the research topic. For instance, in reviewing Apple Inc.'s financial performance in 2021, sections such as Introduction, Company Overview, and Conclusion are identified to organize the report logically.

2. **Section Search.** For each section, the model initially generates $K$ queries aimed at retrieving relevant data and insights. These queries are used to search both local databases and online resources, aggregating the retrieved content. The model then conducts a self-assessment of the gathered information to determine if it is sufficient for drafting the section. If the content is deemed insufficient, additional queries are generated and executed to fill any gaps in information. This iterative process continues until ample data is acquired, allowing the system to proceed to the next stage.

3. **Section Writing.** Utilizing the collected evidence, the system generates detailed analysis texts for each section. This phase includes the integration of tables and charts, enhancing the report's informative quality and visual clarity.

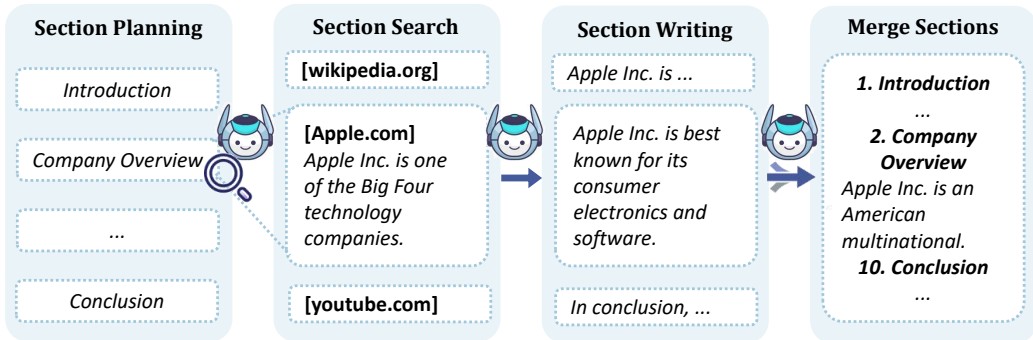

Figure 5: Report composition workflow of topic *Financial performance review of Apple Inc. in 2021*: The four-step process begins with *section planning*, where major section titles are established based on the research topic. This is followed by *section search*, which involves precise queries to gather relevant evidence from local and online sources for each section. *Section writing* utilizes the collected evidence to create detailed analysis texts, complete with tables and charts. Finally, *merge sections* compiles all section analyses into a cohesive report, optimizing narrative flow and outputting a complete document.

4. **Merge Sections.** The final step involves compiling all developed sections into a cohesive report. The system optimizes narrative flow, ensuring that the document presents a comprehensive, coherent analysis of the research topic, concluding with a finalized output ready for dissemination.

## 4 EXPERIMENTS

**Baseline Models.** Baseline LLMs include Claude3.7 (Anthropic, 2023), DeepSeek-R1 (DeepSeek-AI, 2025), DeepSeek-v3 (DeepSeek-AI et al., 2025), GPT4o (OpenAI et al., 2024), and Qwen-72B (Qwen et al., 2025). In addition, we have introduced capacity-enhanced models, such as LongWriter (Bai et al., 2024) and Suri (Pham et al., 2024). LongWriter leverages unique methodologies such as the AgentWrite pipeline to facilitate coherent text generation exceeding 20,000 words. Similarly, Suri employs a multi-constraint instruction-following strategy to generate significantly longer texts while ensuring quality and compliance with constraints.

**Evaluation.** To assess the capabilities of current language models in processing dynamic and real-time data, we employed our newly developed benchmark, DynamicBench. This benchmark surpasses traditional methodologies by focusing on the acquisition and analysis of time-sensitive information. By utilizing web search and database queries, we challenge models to demonstrate their proficiency in handling up-to-date domain-specific queries. This approach provides a comprehensive evaluation of a model's ability to integrate the latest information dynamically and construct accurate reports across various specialized fields. Evaluation dimensions include *accuracy*, *completeness*, *readability*, *applicability*, and *length*.

### 4.1 ABLATION

**Dual-Path Retrieval Ablation.** To validate the effectiveness of our dual-path retrieval design, we conducted ablation experiments comparing online-search-only and local-search-only approaches. Results in Tab. 1 show that removing online retrieval reduces accuracy from 74.8% to 68.2% and applicability from 71.7% to 70.9%, while eliminating local retrieval impairs readability, dropping from 78.0% to 72.1%. Our dual-path approach achieves optimal performance across all metrics.

**Report Generation Pipeline Ablation.** Furthermore, we perform ablation studies to verify the contributions of the planning, retrieval, and fusion modules in our report generation pipeline. The results in Tab. 2 indicate that removing retrieval severely hurts accuracy, dropping from 74.8% to 62.3%; eliminating planning reduces length from 74.4% to 66.8% and completeness from 73.7% to

Table 1: Ablation experiment demonstrating the contribution of local and online retrieval in dual-path design.

| Design | Accuracy | Completeness | Readability | Applicability | Length | Average |
|---|---|---|---|---|---|---|
| w/o Online Retrieval | 74.0 | 67.5 | 72.1 | 71.5 | 72.1 | 71.4 |
| w/o Local Retrieval | 68.2 | 70.3 | 75.6 | 70.9 | 69.8 | 70.9 |
| Ours | **74.8** | **73.7** | **78.0** | **71.7** | **74.4** | **74.5** |

Table 2: Ablation study comparing the contributions of planning, retrieval, and fusion modules.

| Design | Accuracy | Completeness | Readability | Applicability | Length | Average |
|---|---|---|---|---|---|---|
| w/o Retrieval | 62.3 | 68.1 | 78.2 | 66.8 | 73.2 | 69.7 |
| w/o Planning | 73.4 | 67.2 | 76.1 | 71.5 | 66.8 | 71.0 |
| w/o Fusion | 72.6 | 73.4 | 70.5 | 70.1 | 72.5 | 71.8 |
| Ours | **74.8** | **73.7** | **78.0** | **71.7** | **74.4** | **74.5** |

67.2%; and excluding fusion impairs readability (from 78.0% to 70.1%). Our full approach achieves optimal performance, demonstrating how all three components work synergistically together.

## 4.2 RESULTS

In Tab. 3, we present the outcomes of evaluating our method against baseline models under two conditions: **w/o doc** and **with doc**. The **w/o doc** setting involves baseline LLMs responding without the assistance of external documents, while the **with doc** setting allows them to utilize our system's dual-path retrieval results. These settings are for assessment of each model's ability to process information independently versus leveraging additional context.

**LLMs w/o Doc.** Our method demonstrated new state-of-the-art performance across all dimensions. In terms of *accuracy*, our model achieved 74.8%, outperforming GPT4o by 16.6%. The current SOTA in this category was the capability-enhanced model LongWriter, which reached 69.3%. For *completeness*, our approach attained a score of 73.7%, exceeding DeepSeek-v3 by 8.1%. When evaluating *readability*, Claude3.7-Sonnet excelled with a score of 78.7%, closely followed by our model, which scored 78.0%. In terms of *applicability*, our model demonstrated a score of 71.7%, slightly surpassing Claude3.7-Sonnet. Regarding *length*, our model surpassed the competition with a score of 74.4%, outperforming the next best model, GPT4o, by 7.4%. Across the five dimensions, our method achieved an average score of 74.5%, surpassing the current SOTA GPT4o by 7.0%.

**LLMs with Doc.** With access to relevant documents, our method continued to showcase state-of-the-art performance across all evaluated metrics. In terms of *accuracy*, our model achieved an impressive 74.8%, outperforming current SOTA Claude3.7-Sonnet by 5.5%. For *completeness*, our approach scored 73.7%, exceeding DeepSeek-v3's score by 4.3%. Claude3.7-Sonnet led the performance in *readability* with a score of 78.7%, with our model closely following at 78.0%. Our model demonstrated superior *applicability*, scoring 71.7%, which slightly surpassed Claude3.7-Sonnet. In terms of *length*, our model excelled with a score of 74.4%, significantly outperforming DeepSeek-v3, which scored by 6.1%. Overall, across the five dimensions, our method achieved an average score of 74.5%, substantially higher than Claude3.7-Sonnet and GPT4o by 4.2% and 5.8%.

**Comparison.** The evaluation of models with and without document access reveals notable differences in performance. For general LLMs such as Qwen2.5-72B-Instruct, DeepSeek-v3, GPT4o, and Claude3.7-Sonnet, the performance generally improved significantly when relevant documents were provided. This enhancement highlights LLMs' capacity to leverage external context effectively. Conversely, for capability-enhanced models like Suri and LongWriter, a decline in performance was observed with the inclusion of document. This suggests that these models, which are optimized for generating extended text, may sacrifice some ability to comprehend long contexts when supplied with additional documents. The tendency may result in decreased readability and completeness when external data is introduced. Moreover, both DeepSeek-v3 and Suri, which involve reinforce-

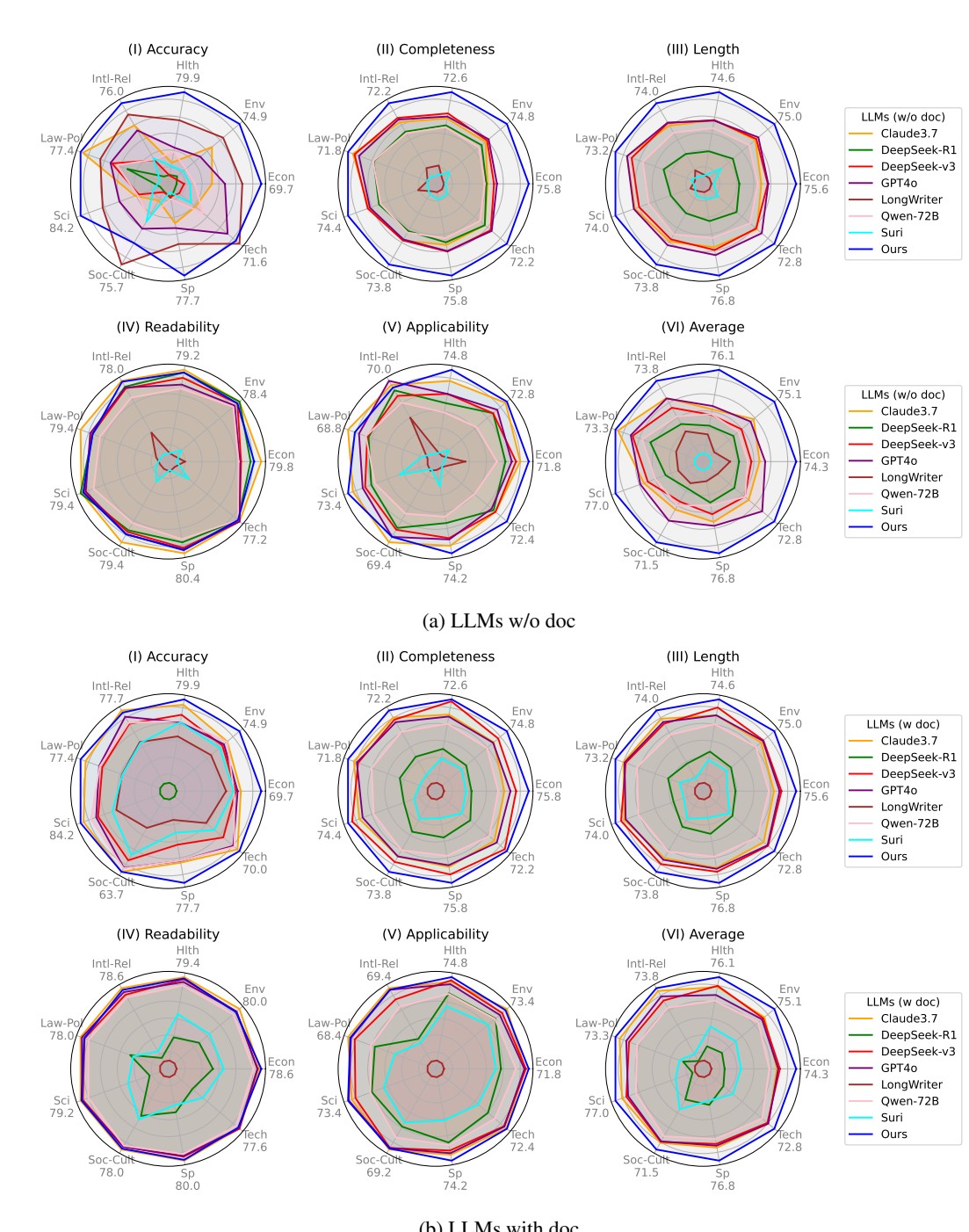

(a) LLMs w/o doc

(b) LLMs with doc

Figure 6: Performance of various Systems and LLMs includes Claude3.7 (Anthropic, 2023), DeepSeek-R1 (Anthropic, 2023), DeepSeek-v3 (DeepSeek-AI et al., 2025), GPT4o (OpenAI et al., 2024), LongWriter (Bai et al., 2024), Qwen-72B (Qwen et al., 2025), Suri (Pham et al., 2024). Evaluation metrics include accuracy, completeness, length, readability, applicability, and average performance. Each model is assessed using a range of topics, such as economy (Econ), environment (Env), health (Hlth), international relations (Intl-Rel), law and politics (Law-Pol), science (Sci), society and culture (Soc-Cult), sports (Sp), and technology (Tech).

Table 3: Evaluation metrics for LLMs encompassing various aspects, including Accuracy (Acc), Completeness (Comp), Readability (Read), Applicability (App), and Length (Len). With Doc and w/o Doc indicate whether the models were provided with relevant documents. The best and second-best results are highlighted using bold and underlined formatting.

| Models | Acc. | Comp. | Read. | App. | Len. | Average |
|---|---|---|---|---|---|---|
| **w/o Doc** | | | | | | |
| DeepSeek-R1 (DeepSeek-AI, 2025) | 40.8 | 62.0 | 77.6 | 69.3 | 52.3 | 60.4 |
| DeepSeek-v3 (DeepSeek-AI et al., 2025) | 44.1 | 65.9 | 77.4 | 69.9 | 64.6 | 64.4 |
| Qwen2.5-72B-Instruct (Qwen et al., 2025) | 49.3 | 61.3 | 75.8 | 68.1 | 60.8 | 63.1 |
| GPT4o (OpenAI et al., 2024) | 58.2 | 65.7 | 77.3 | 70.4 | 66.0 | 67.5 |
| Claude3.7-Sonnet (Anthropic, 2023) | 55.0 | 64.7 | **79.0** | 71.3 | 64.5 | 66.9 |
| Suri (Pham et al., 2024) | 43.9 | 45.5 | 62.6 | 63.0 | 43.0 | 51.6 |
| LongWriter (Bai et al., 2024) | 68.0 | 45.4 | 62.4 | 62.8 | 41.5 | 56.0 |
| **with Doc** | | | | | | |
| DeepSeek-R1 (DeepSeek-AI, 2025) | 15.3 | 47.0 | 53.0 | 64.4 | 42.9 | 44.5 |
| DeepSeek-v3 (DeepSeek-AI et al., 2025) | 59.9 | 69.4 | 77.2 | 70.1 | 68.3 | 69.0 |
| Qwen2.5-72B-Instruct (Qwen et al., 2025) | 63.6 | 60.3 | 75.4 | 66.8 | 60.4 | 65.3 |
| GPT4o (OpenAI et al., 2024) | 63.4 | 65.6 | 77.3 | 70.5 | 67.0 | 68.7 |
| Claude3.7-Sonnet (Anthropic, 2023) | 69.3 | 65.9 | **78.7** | 71.2 | 66.3 | 70.3 |
| Suri (Pham et al., 2024) | 51.7 | 40.2 | 57.2 | 60.9 | 37.2 | 49.5 |
| LongWriter (Bai et al., 2024) | 45.0 | 30.1 | 40.2 | 47.1 | 26.8 | 37.8 |
| Ours | **74.8** | **73.7** | 78.0 | **71.7** | **74.4** | **74.5** |

ment learning through human feedback (RLHF) for fine-tuning, exhibited this pattern, indicating that their training methodologies might prioritize generative aspects over contextual understanding.

### 4.3 CATEGORY-LEVEL ANALYSIS

In Fig. 6, we present the detailed results of LLMs and systems across various domains. For **LLMs w/o doc**, although the average results of previous methods differ significantly from ours, in some domains, better results can be achieved. For example, LongWriter shows slightly higher accuracy in the fields of *Society & Culture* and *Technology* than ours, and Claude3.7 has slightly better applicability in *Law & Politics* and *Society & Culture*. A possible reason for this is that these models develop preferences during training, possibly due to the inclusion of specific knowledge not contained in web searches or the available databases. It is evident that the average results of **LLMs with doc** show significant improvement compared to **LLMs w/o doc**, stemming from the supplementary external information enhancing the inherent knowledge of LLMs. While the results of **LLMs with doc** have narrowed the gap with our system, they generally do not surpass our system, which can be attributed to the fact that both **LLMs with doc** and our system utilize the same dual-path retrieval information. However, our method effectively leverages this information through a systematic approach.

## 5 CONCLUSION

This work presents advancements over traditional benchmarks for evaluating large language models (LLMs) by introducing DynamicBench, a dynamic benchmark developed to assess real-time information acquisition and processing capabilities. Utilizing a dual-path retrieval system that synergizes local report databases with web searches, DynamicBench offers comprehensive and objective evaluations across diverse domains. This benchmark demands models to demonstrate domain-specific knowledge, ensuring the generation of accurate reports. Additionally, we have developed an advanced report generation system capable of managing the complexities inherent in dynamic information synthesis. Through systematic planning, query generation, and resource aggregation, this system integrates up-to-date information to produce detailed, coherent reports reflecting the latest data trends. Our experimental results underscore its effectiveness, demonstrating state-of-the-art performance that exceeds existing models like GPT4o across various scenarios.

## LIMITATIONS

While DynamicBench represents an advancement in the evaluation of LLMs by incorporating real-time data retrieval and processing, several limitations remain. Firstly, the dependency on both web searches and local report databases means that the benchmark's effectiveness is contingent on the quality and accessibility of these external sources. Discrepancies or biases in the available data can potentially affect the accuracy and objectivity of the evaluation results. Additionally, the scope of documents considered might not capture the full breadth of contextual knowledge required for specialized fields. The benchmark may not fully assess the depth of understanding necessary for niche domains that require highly specific insights and expertise. These limitations highlight areas for potential improvement, paving the way for future work focused on enhancing data integration strategies, and expanding domain coverage to further advance LLM evaluation and report generation methodologies.

## BROADER IMPACT

As AI models become increasingly capable of handling real-time information, there are considerations surrounding the ethical use and potential misuse of these technologies. The ability to rapidly generate detailed, coherent reports and real-time data integrations increases the risk of deploying LLMs for misleading or biased content creation. Researchers and developers must prioritize the mitigation of such risks. By fostering transparency and accountability in AI practices, we can ensure that the positive impacts of our work are realized while curtailing the possibilities for harm or misuse. Ultimately, our efforts aim to empower stakeholders with enhanced tools for navigating the complexities of modern information landscapes responsibly.

## AI ASSISTANCE DISCLOSURE

The authors incorporated LLMs to aid in drafting sections of this manuscript. After the initial creation of the text, the authors thoroughly reviewed and refined the material, ensuring its accuracy and integrity, and they assume complete responsibility for the published work.

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

## A  PROMPT TEMPLATE

Table 4: Complete prompt templates for financial analysis tasks

| Template Type | Prompt Template |
|---|---|
| Adding documents into baseline models | You are a Chief Financial Analyst tasked with researching the topic: {query}. Your goal is to write a comprehensive and detailed analysis of this theme in English. If there are numerous numbers or if a visual representation would better convey the information, please create a chart. For tables, use Markdown format. For more complex charts like line charts or bar charts, please use SVG code to generate them. The output should be formatted in HTML, making sure that all tables and SVG charts are properly displayed and rendered. You should pay attention to the SVG code, ensuring the charts are properly displayed. Please use ¡section¿ HTML tags to divide the sections. Here are the relevant documents for the topic report: {doc} |
| Planning | As a financial analyst, your task is to generate a table of contents for a research report based on the given research topic, including the main chapters (level one headings). The output format should be a Python list, providing the main components of the report, such as: ['section1', 'section2', 'section3']. Only generate the main sections, excluding appendices, executive summaries, references, etc. Do not include any subsections or further details. |

Table 4: Complete prompt templates for financial analysis tasks (continued)

| Template Type | Prompt Template |
|---|---|
| Generating query list | You are a chief financial analyst. The research topic is {args.topic}. Your task is to gather evidence for the section {section}. The following is the information you already have: {available_information}. First, analyze if the above information is sufficient for this section. You should focus on the numbers and ensure their comprehensiveness. If the information is sufficient, output only 'Sufficient information found', nothing else. If the information is insufficient, generate up to 5 precise search queries to help collect comprehensive information for this section. The output format should be a Python list, e.g., ['query1', 'query2', 'query3']. |
| Writing sections | You are a chief financial analyst, and your task is to research the topic: {args.topic}. The following is the evidence for section {section}:\n {doc}. Your goal is to write a comprehensive and detailed analysis of this section, using English. If there are many numbers or graphical representations that can better convey the information, please create charts. Use Markdown format for tables. For more complex charts, such as line charts or bar charts, use SVG code to generate them. The output format should be HTML, ensuring that all tables and SVG charts are correctly displayed and rendered. Please ensure the provided SVG code makes the charts display correctly. Please enclose the entire content of this section in ¡section¿ HTML tags. |
| Merging sections | You are a chief financial analyst, tasked with combining and improving the provided section summaries: {sections}. The report's title is {args.topic}. Please carefully refine each section to make the language clear and professional. Once each part is perfected, integrate them into a comprehensive and coherent financial report. The report should be output in HTML format, ensuring that all tables and SVG charts are correctly displayed and rendered. No additional comments or explanations are needed; please provide only the HTML code. |
| Extracting evaluation queries | Please extract key information from the following report and generate Q&A pairs. Use newline to connect queries, output up to 10 Q&A pairs, and do not output additional content. Example: 
 user: Tenergy Technology released its 2024 report on March 21: In 2024, the company achieved revenue of 4.45 billion yuan, with a year-on-year growth of 30.96%. 
 assistant: Q: What was Tenergy Technology's revenue in billion yuan for 2024? A: 4.45 billion yuan 
 Q: What was the year-on-year growth percentage for Tenergy Technology's revenue in 2024? A: 30.96% 
 {section} |

Table 4: Complete prompt templates for financial analysis tasks (continued)

| Template Type | Prompt Template |
|---|---|
| Scoring Accuracy | Your task is to evaluate the accuracy of a research report. Please rate it from 1 to 10 based on the following criteria, and provide a brief explanation to help the author understand the basis of your rating and suggestions for improvement. Rating Criteria: - 1-2 points: Severely inaccurate. Significant errors or misleading information are present in the report. - 3-4 points: Inaccurate. Multiple errors or inaccuracies affect the overall credibility of the report. - 5-6 points: Basically accurate. The report is generally accurate, but some data or conclusions may need verification. - 7-8 points: Fairly accurate. The report content is mostly accurate, with only minor corrections needed. - 9-10 points: Fully accurate. The report content is entirely correct with no apparent errors. Specific Requirements: - Evaluate whether the report's data and conclusions are accurate and consistent with the provided references. - Ensure the research report cites appropriate references to support its conclusions. - Consider whether the report's analysis and explanations are thorough and support its accuracy. Title: {query} Content: {section} References: {searched_results} |
| Scoring Completeness | Your task is to evaluate the completeness of a research report. Please rate it from 1 to 10 based on the following criteria, and provide a brief explanation to help the author understand the basis of your rating and suggestions for improvement. Rating Criteria: - 1-2 points: Very incomplete. The report lacks the majority of necessary elements. - 3-4 points: Incomplete. The report contains only some basic elements. - 5-6 points: Basically complete. The report covers most necessary parts but requires considerable improvement. - 7-8 points: Quite complete. The report is well-organized with only minor optimizations needed. - 9-10 points: Complete. The report covers all necessary parts without need for improvement. Specific Requirements: - Evaluate whether the report includes all necessary components. - Consider the depth of analysis and whether the report adequately addresses its objectives. Title: {query} Content: {section} |
| Scoring Readability | Your task is to evaluate the language and expression of a research report. Rate the report from 1 to 10 according to the following criteria, and provide a brief explanation to help the author understand the basis of the rating and suggestions for improvement. Criteria: - 1-2 points: Poor language expression. The report contains significant language errors, making it difficult to understand. - 3-4 points: Average expression. Some language in the report is unclear or inaccurate. - 5-6 points: Generally good. The report's language is fairly clear, but there's room for improvement. - 7-8 points: Clear expression. The report's language is smooth and accurate. - 9-10 points: Excellent language. The report uses precise and clear language to convey content, no improvements needed. Specific requirements: - Evaluate the clarity and professionalism of the language. - Check for any grammar or spelling errors, and assess if professional terminology is used appropriately. Title: {query} Content: {section} References: {searched_results} |

Table 4: Complete prompt templates for financial analysis tasks (continued)

| Template Type | Prompt Template |
|---|---|
| Scoring Length | Your task is to assess whether the length of a research report is sufficient. Please rate it from 1 to 10 based on the following criteria, and provide a brief explanation to help the author understand the basis of your rating and suggestions for improvement. Rating Criteria: - 1-2 points: Highly insufficient length. The report is too short to effectively convey information. - 3-4 points: Insufficient. The report length is inadequate, affecting the delivery of important information. - 5-6 points: Basically sufficient. The report length conveys core information but needs expansion to include more details. - 7-8 points: Fairly sufficient. The report length effectively conveys information but has room for improvement. - 9-10 points: Sufficient. The report length is just right, effectively covering all necessary information. 

 Specific Requirements: - Evaluate whether the report length is sufficient, considering the complexity and richness of the information. - Assess whether each chapter contains adequate information, and whether the number of chapters is sufficient to reflect all aspects of the title. - Ensure that the content of the report matches its length and chapter arrangement, aiming for comprehensive and detailed coverage. 
 Title: {query} Content: {section} |
| Scoring Applicability | Your task is to evaluate the applicability of a research report. Rate the report from 1 to 10 according to the following criteria, and provide a brief explanation to help the author understand the basis of the rating and suggestions for improvement. 

 Criteria: - 1-2 points: Poor applicability. The research does not contribute to solving any practical problems. - 3-4 points: Low applicability. The research provides limited assistance to practice. - 5-6 points: Generally applicable. The research can be applied to practical issues to some extent. - 7-8 points: Highly applicable. The research can be well-applied in practice. - 9-10 points: Very applicable. The research findings have broad and significant application value. 
 Specific requirements: - Evaluate whether the research can be directly applied to real-world problems. - Consider the practical application value and potential of the research results. 
 Title: {query} Content: {section} References: {searched_results} |

## B  EXAMPLES OF SCORING CRITERIA

Table 5: Examples of scoring criteria across evaluation metrics

| Example Content |
|---|
| 1 point Length and Completeness |

```
<svg width="800" height="400" xmlns="http://www.w3.org/2000/svg">
  <rect width="100" height="200" x="50" y="180" fill="#4CAF50" />
  <rect width="100" height="260" x="200" y="120" fill="#2196F3" />
  <rect width="100" height="160" x="350" y="220" fill="#FFC107" />
  <rect width="100" height="280" x="500" y="100" fill="#F44336" />

  <text x="70" y="380" font-size="16" fill="#333">Automation</text>
  <text x="220" y="380" font-size="16" fill="#333">Collaboration</text>
  <text x="370" y="380" font-size="16" fill="#333">AI Features</text>
  <text x="520" y="380" font-size="16" fill="#333">Cloud Scalability</text>
</svg>
```

Table 5: Examples of scoring criteria across evaluation metrics (continued)

---

**Example Content**

---

9 point Applicability

```
<section>
    <h2>Recommendations by Population Groups</h2>
    <table>
        <thead>
            <tr>
                <th>Population Group</th>
                <th>Physical Activity Recommendation</th>
            </tr>
        </thead>
        <tbody>
            <tr>
                <td>Children and Adolescents (5{17 years)</td>
                <td>At least an average of 60 minutes per day of
                moderate-to-vigorous intensity aerobic physical
                activity across the week. Include vigorous activities
                and activities that strengthen muscle and bone at
                least 3 days per week.</td>
            </tr>
            <tr>
                <td>Adults (18{64 years)</td>
                <td>150{300 minutes of moderate-intensity aerobic
                physical activity per week, or 75{150 minutes of
                vigorous-intensity aerobic physical activity, or an
                equivalent combination. Muscle-strengthening activities
                on 2 or more days per week.</td>
            </tr>
            <tr>
                <td>Older Adults (65+ years)</td>
                <td>Same as adults, but additionally activities that
                emphasize balance and functional training to prevent falls.</td>
            </tr>
            <tr>
                <td>Pregnant and Postpartum Women</td>
                <td>At least 150 minutes of moderate-intensity aerobic
                physical activity per week. Incorporate muscle-strengthening
                activities as appropriate.</td>
            </tr>
            <tr>
                <td>People with Chronic Conditions or Disabilities</td>
                <td>Engage in physical activities according to abilities:
                150{300 minutes of moderate-intensity aerobic activity,
                strengthening exercises, and balance training when appropriate.</td>
            </tr>
        </tbody>
    </table>
</section>
```

---

1-point Applicability (Excerpt)

```
<section>
    <h2> "AT&T Expands Leadership Role in Smart Cities" </h2>
    <p> The "AT&T Expands Leadership Role in Smart Cities" report,
    published on September 10, 2015, highlights AT&T's commitment
    to building smart cities at scale and helping communities change
    the game by connecting things like utility meters, streetlights,
    and water systems. The report mentions that AT&T's work in Chicago
    and plans to build a smart city in North Carolina, aims to simplify
    connections in complex systems and help cities save money, conserve
    energy, improve public safety, and further engage with their citizens.
    The report states that cities cannot realize their potential for
    economic growth and public safety without secure connectivity and
    the ability to gain insights from data, permitting them to identify
    and address problems in real-time. The report calls smarter cities
    the future of urban infrastructure and mentions that it's an intelligent
    infrastructure that does more than just modernizing a city; it makes
    it analyze, adapt, and continually improve. The report notes that from
    water system sensors to advanced analytics and energy efficiency efforts,
    cities are becoming smarter. The report mentions that city officials are
    using data to improve bus and train schedules, helping drivers avoid
    traffic, and even improving parking enforcement, which is good news for
    commuters, citizens, and tourists. The report concludes by saying that
    the future of smarter cities is here, and the possibilities are endless. </p>
</section>
```

---

Table 5: Examples of scoring criteria across evaluation metrics (continued)

**Example Content**

9-point Readability (Excerpt)

```html
<!DOCTYPE html>
<html lang="en">
<head>
    <meta charset="UTF-8">
    <meta name="viewport" content="width=device-width, initial-scale=1.0">
    <title>Huawei's Development of 5G Infrastructure Technology and Its Global Impact</title>
</head>
<body>
    <header>
        <h1>Huawei's Development of 5G Infrastructure Technology and Its Global Impact</h1>
        <p>Comprehensive analysis of Huawei's role in 5G infrastructure development
        and the resulting global implications.</p>
    </header>
    <section id="introduction">
        <h2>Introduction</h2>
        <p>The development of 5G technology has been at the forefront of modern
        telecommunications, promising higher speeds, lower latency, and increased connectivity.
        Huawei, a leading global telecommunications company, has emerged as a significant
        player in the 5G industry.
        This analysis explores Huawei's strategic investments in 5G technology,
        innovations, global influence, and the geopolitical and economic impacts
        of its advancements.</p>
    </section>
    <section id="investment-and-innovations">
        <h2>Investment and Innovations in 5G Technology</h2>
        <p>Huawei's commitment to upgrading global telecom infrastructure is evident through
        substantial investments. Over the past decade, Huawei has allocated more than
        $4 billion toward 5G research and development,
        leading to advancements in Industry 4.0 industries such as AI-driven quality control
        and self-driving vehicles.
        Below is a table illustrating Huawei's R&D investment in 5G technologies
        compared to other companies:</p>
...
    <footer>
        <p>© 2024 | Chief Financial Analyst Report | All Rights Reserved</p>
    </footer>
</body>
</html>
```

