# OpenReview forum: "DynamicBench: Evaluating Real-Time Report Generation in Large Language Models"
_ICLR.cc/2026/Conference — ICLR 2026 Conference Withdrawn Submission_

### Official Review · Reviewer_9nzi · 2025-10-23

**Soundness:** 2
**Presentation:** 3
**Contribution:** 2
**Rating:** 4
**Confidence:** 3

**Summary:**

This paper present DynamicBench to evaluate the ability of LLM to acquire, process and synthesize real-time information  for report generation. It integrated dual-path retrieval from web and local databases.

**Strengths:**

- The benchmark spans multiple specialized fields (Tech & Science, Economy & Environment, Culture & Health, International & Politics)
- The four-stage process (planning, search, writing, merging) ensures structured, accurate, and coherent outputs

**Weaknesses:**

- This paper emphasis the importance of real-time data but rely heavily on the web and local databases. Local database are collected from AnnualReport, which is also not real-time data.
- Are you planing to release the benchmark?
- Despite the amazing performance in Figure6, can you make sure that the comparison is fair? In your methods, you did a multi-stage workflow including planning, search, writing and merge. What about other methods? Are they writing report end2end? If so, you should select some other agent methods for comparison.
- THere should be a comprehensive comparison with other benchmarks like LongWriter to show your uniqueness. Now, I can not tell the novelty of this benchmark.

**Questions:**

How does DynamicBench handle conflicting or outdated information retrieved from web and local sources, and are there mechanisms for bias detection?

Please fix the typos in Table4

---

### Official Review · Reviewer_bgCf · 2025-10-29

**Soundness:** 1
**Presentation:** 1
**Contribution:** 1
**Rating:** 0
**Confidence:** 5

**Summary:**

This paper proposes a benchmark called DynamicBench to measure LLMs' ability to generate reports with up-to-date information. It also designs a four-step report generation inference pipeline and shows its empirical performance.

**Strengths:**

1. The topic itself (evaluating LLMs' ability to generate reports based on real-time / up-to-date information) is highly practical and relevant to real-world usage of LLMs, so this research direction is worth pursuing.

2. The benchmark includes data from a variety of categories including technology & science, economy & environment, culture & health, and internation & politics.

3. Experiments are done with different LLMs and abalations.

**Weaknesses:**

1. Based on the contents of the paper, my major concern is that the design of DynamicBench does not really measure the "real-time" or "up-to-date" report generation ability in LLMs as the authors claimed. Take one of the query examples from Figure 1, "The growth and challenges of the global semiconductor industry from 2024 to 2025", this query will be outdated in the years after 2025. Also, the authors didn't provide details about how the input queries (i.e., the queries used to ask LLMs for generating reports) are collected, so readers cannot be sure how the benchmark design necessitates LLMs to process up-to-date information in order to generate reports. (The lack of details also bring us to the next point)


2. This paper lacks important details in the following aspects:

(1) Dataset construction: How are the input queries (e.g., those shown in Figure 1) collected? Even the size of the dataset wasn't provided.

(2) Evaluation process: The paper only describes the five metrics used to evaluate report quality in subsection 3.1.2, but doesn't state the following

a. Which model(s) are used as evaluators to grade the input report with these five metrics?

b. How good is the evaluation itself? Is these any evidence that this evaluation process is reasonable and correlates well with report quality?

3. There is still a large room for improvement in terms of the writing and presentation of this paper. Initially, I had a very hard time understanding what the authors tried to evaluate, and precisely how this work differs from existing works. In order to understand the paper, I have to go back and forth and re-reading the paper multiple times.

To summarize, there are two major flaws in this work. First, the design of DynamicBench doesn't support the authors' claim that it measure LLMs' real-time report generation abilities, at least from the current contents in the paper. Second, the writing is not clear and the paper lacks many important details in benchmark construction and evaluation process. I believe this work is highly incomplete and requires substantial improvements.

**Questions:**

1. How is your input data (i.e., query examples listed in Figure 1) collected? What's the size of the dataset?

2. How do you ensure that DynamicBench evaluate "real-time" report generation ability?

3. How good are your evaluation criteria and evaluation process? Do they actually approximate report quality well enough? What's the model used as the evaluator? For example, do you use LLM-as-a-judge?

---

### Official Review · Reviewer_ymRP · 2025-10-31

**Soundness:** 4
**Presentation:** 4
**Contribution:** 3
**Rating:** 6
**Confidence:** 5

**Summary:**

This paper presents DynamicBench, which is a new benchmark for evaluating how well large language models (LLMs) can handle real-time information and generate reports from it. The authors say that most current benchmarks are too static (they just use storytelling and opinion stuff), so they don’t really test what LLMs need for actual real-world tasks. Here, DynamicBench uses both live web searches and local report databases to see how the models perform when they get fresh info or have extra documents for context. The results look pretty good. apparently their approach does better than GPT4o on some key measures in both document-free and document-assisted scenarios.

I think the claims are well-supported; the setup makes sense, and experiments seem thorough. They compare against a strong baseline (GPT4o) and do a bunch of studies to show why their approach works.

The writing is clear for the most part, and figures/tables help explain what’s going on. I didn’t feel lost while reading, and it’s easy to see how this fits with older benchmarks and why it’s needed now.

**Strengths:**

It’s a fresh idea, benchmarking for dynamic info rather than static stuff.The way they combine web and database searching seems practical.Experiments are comprehensive, lots of different evaluation criteria. Results sound impressive, and they tackle a real gap in current LLM evaluation.

**Weaknesses:**

Hard to say how well this works for real niche domains (like legal/medical info)—could be limited by what’s on the web or in their databases. If the external data sources are bad or biased, the benchmarks could be off. I’m not sure how robust it is against bad info. Some models with longer outputs seemed less readable/comprehensible—curious if that’s a fixable issue or just a tradeoff. Even with rating guidelines, measures like “readability” and “applicability” still have some subjectivity. maybe more standardization could help.

**Questions:**

How does DynamicBench handle with web data that’s wrong, incomplete, or just plain adversarial? How can DynamicBench get regular updates as sources and domains change? that doesn't seem to be explained in detail.

---

### Official Review · Reviewer_vhrT · 2025-11-01

**Soundness:** 1
**Presentation:** 2
**Contribution:** 1
**Rating:** 2
**Confidence:** 5

**Summary:**

This paper introduces DynamicBench, a benchmark designed to evaluate LLMs’ ability to generate reports by accessing and processing real-time information. It tests whether models can access and utilize up-to-date knowledge through web searches and database queries for accurate responses. Using a dual-path retrieval pipeline that combines local report databases with online searches, DynamicBench assesses models in both with and without document assistance scenarios. Beyond the benchmark, the paper presents a report generation system that plans sections, retrieves relevant data, and self-assesses the need for additional information. Experimental results demonstrate that this approach achieves state-of-the-art performance.

**Strengths:**

1. The paper is clearly written with informative figures and structure.
2. The topic is timely and highly relevant to current LLM capabilities (real-time information retrieval, and report generation).
3. The overall idea is easy to understand.

**Weaknesses:**

1. **Reliability of the accuracy evaluation**
   My main concern is that the proposed accuracy evaluation methodology lacks sufficient rigor. Several related issues are discussed below:

   a. **Error propagation in intermediate steps**
      The paper evaluates report accuracy by generating QA pairs from the input report (apparently using an LLM), then querying both a local report database and the web to verify answers. However, these intermediate steps also introduce potential noise. For example, how do we ensure that the generated QA pairs faithfully represent the report content? The authors should consider strategies such as using multiple strong LLMs and majority voting to reduce this uncertainty. Moreover, since the generated reports may contain markdown tables or SVG charts, extracting correct answers from such complex formats is non-trivial, raising further concerns about evaluation reliability.

   b. **Coverage of QA pairs**
      The prompt limits QA generation to “up to 10 QA pairs.” There is no evidence that these 10 questions adequately capture the full informational scope of the report, especially when reports may contain many factual statements, numerical tables, or charts. The authors do not describe any mechanism to ensure random or representative sampling. It is possible that the generated questions focus only on the simplest or most salient content, ignoring information buried in tables, charts, and later sections.

   c. **Ambiguity and answer simplicity**
      The examples shown are short and unambiguous, but how does the system ensure this property generally? What if a question involves multi-year trends or complex quantitative data? The provided prompts seem insufficient to guarantee unambiguous QA pairs. Even if all generated QAs are simple, the limited coverage issue remains: ten short questions cannot fully represent report-level accuracy.

   d. **Inconsistency in information sources**
      The system allows both local retrieval and online search. It is unclear whether the online sources used during report generation and evaluation are the same. If the sources differ, discrepancies might arise from different retrieval results rather than from the model’s factual errors. The large performance gap between “without local retrieval” and “without online retrieval” in the ablation study might partially stem from this inconsistency.

2. **Evaluation metrics beyond accuracy**
   Apart from the accuracy metric, all other evaluations appear to rely on prompt-based LLM self-ratings, which are inherently subjective and potentially unstable. It is unclear whether the same model would produce consistent scores across multiple runs, or whether score differences (e.g., between 7 and 8, or 9 and 10) are statistically meaningful. The paper provides only very limited qualitative examples of the LLM-based evaluations, focusing on extreme scores (e.g., 1 vs. 10), without examining intermediate ratings or assessing their alignment with human judgment.

3. **Unclear scope of “report generation”**
   The scope of the reports considered in this work is not clear. Report generation is a broad task, particularly for data-centric reports that contain rich design elements such as tables and charts. While the prompts reference markdown tables and SVG charts, and the provided examples include web-style structures (e.g., footers), it is unclear how these elements are handled in the benchmark. The authors should explicitly define the scope of reports considered in DynamicBench and clarify how various stylistic or structural features (tables, charts, footers, etc.) impact evaluation. For instance, does the presence of a footer or chart influence model scores? Are chart rendering and table readability incorporated into the evaluation metrics? It also appears that potential display or formatting errors in tables and charts are not accounted for in the metrics. Given that these elements are common and challenging, the benchmark should address them in both the evaluation framework and analysis.

4. **Lack of novelty**
   The paper positions DynamicBench as its core contribution and the accuracy evaluation of real-time information retrieval as the main technical idea. However, as noted above, the accuracy evaluation methodology lacks reliability. Beyond this, the benchmark itself is relatively simple and does not incorporate sufficient mechanisms to ensure rigor or to account for the broad scope of report types. Similarly, the proposed report generation approach is straightforward and direct, offering limited methodological novelty or conceptual advancement.

**Questions:**

1. When generating reports, does each query retrieve a single local document or multiple documents per section (e.g., one per query)? For the baselines that generate reports directly, are the same documents provided? Clarifying this is essential for fairness.
2. In the evaluation stage, the QA pairs are generated using an LLM as well. What base model was used for this step?
3. The accuracy metric mentions that if “relevant content is missing or conflicts are detected,” it may be marked as Cannot Determine or Incorrect. How are Cannot Determine cases treated in accuracy computation?
4. How many reports were generated for evaluation, and how were the evaluation instances selected?

---

### Note · Authors · 2026-01-14

I have read and agree with the venue's withdrawal policy on behalf of myself and my co-authors.